# Mapping Thriving at Work as a Growing Concept: Review and Directions for Future Studies

Ghulam Abid [1] and Francoise Contreras [2,*]

1   Department of Business Studies, Kinnaird College for Women, Lahore 54000, Pakistan
2   School of Management and Business, Universidad del Rosario, Bogota 110141, Colombia
*   Correspondence: francoise.contreras@urosario.edu.co

**Abstract:** This study aims to provide a bibliometric analysis of the literature on thriving at work in psychology and business/management produced between 2001 and 2021, using the Web of Science (WoS) database. The analyses allowed us to identify, through 190 documents, the emergence of the concept of thriving at work and its development. The main research variables related to this concept and its methodology were identified. Likewise, the most influential authors, the most cited articles, the more frequently cited journals, and the countries contributing to developing this construct are analyzed. In addition, an analysis of co-citation, co-occurrences, and bibliographic coupling was conducted. Finally, content analysis of the most popular keywords and the co-citation of cited references are conducted. These analyses allow the identification of the main developments in the topic of thriving at work. The theoretical and practical implications of this bibliometric analysis are discussed.

**Keywords:** thriving at work; social sustainability; bibliometric studies; bibliometric mapping; VOSviewers





## 1. Introduction

Organizations worldwide are experiencing a highly unpredictable and complex business environment that brings uncountable opportunities to gain sustainable competitive advantage and sustainability [1]. It is quite possible to instill a favorable psychological climate at work and promote workplace well-being [2]. With the growing importance of promoting employee well-being and innate growth and development in workplace settings, researchers have been motivated to examine the value of thriving at work. Thriving at work is a "psychological state in which an individual experiences the sense of learning and vitality" [3]. Learning denotes the acquisition and application of new knowledge and skills, whereas vitality represents the positive feelings at work associated with zest and energy. Both dimensions work in tandem to nurture the experience of growth and development to shape a healthy workplace [4] and sustainable performance [5]. The increasing emphasis on workplace thriving is mainly shaped by the view of self-adaptation, stating that a thriving workforce can monitor the health and status of their progress, and in that way make adaptive choices to further their development and subjective well-being [5]. In support of this contention, scholars [6] stated that integrating the sense of employee thriving serves as an essential step to building a positive workplace where individuals' well-being is sustained. Employee well-being has been recognized as an essential component of the employees and an organization's sustainable performance [7].

Thriving at the workplace is an emerging phenomenon; the current literature demonstrates the antecedents and the consequences of thriving. The following are the few well-known antecedents of thriving: perceived organizational support, proactive personality, servant leadership, unit contextual features and resources, transformational leadership, managerial coaching, fairness perception, trust, high-performance work systems, etc. (see Table 1).

**Table 1.** Antecedents and consequences of thriving at work.

| Antecedents of Thriving at Work | | Consequences of Thriving at Work | |
|---|---|---|---|
| Antecedents | Sources | Consequences | Sources |
| Perceived organizational support | [8] | Career adaptability | [9] |
| Heedful relating | [8] | Happiness at work | [10] |
| Psychological contract breach | [10] | OCB | [11] |
| Workplace spirituality | [12] | Work engagement | [13] |
| Knowledge hiding | [14] | Life satisfaction | [15] |
| Proactive personality | [9] | Innovative work behavior | [8] |
| Workplace violence | [16] | Helping behavior | [17] |
| Servant leadership | [18] | Proactivity | [19] |
| Core self-evaluations | [18] | Task performance | [17] |
| Workplace incivility | [20] | Task mastery | [19] |
| Work-family enrichment | [21] | Career commitment | [22] |
| Hindrance stressors | [23] | Affective commitment | [24] |
| Communication and centrality; role ambiguity; role overload | [25] | Career satisfaction | [22] |
| Negative appraisal | [11] | Development | [26] |
| Unit contextual features and Resources | [3] | Career engagement | [22] |
| Transformational leadership | [19] | Career development initiatives | [27] |
| LMX | [28] | Task performance | [27] |
| Authentic leadership | [29] | Voice behavior | [30] |
| Team justice | [31] | Job satisfaction | [32] |
| Managerial coaching; fairness perception; trust | [24] | Burnout | [33] |
| Overall job autonomy; task identity | [32] | Positive health | [18] |
| Supervisor and coworker support | [15] | Turnover intention | [34] |
| Psychological safety | [17] | | |
| Civility; compassion | [35] | | |
| Empowering leadership | [36] | | |
| Psychological climate, support climate | [26] | | |
| High performance work systems | [14] | | |
| Positive affect, negative affect | [27] | | |
| Psychological contract fulfillment | [34] | | |
| Trust in colleagues, supervisor, and management | [37] | | |

Empirical studies on thriving at work have concluded various consequences which are: career adaptability, turnover intention, positive health, job satisfaction, task performance, affective commitment, task performance, helping behavior, happiness at work, work engagement, life satisfaction, etc. (see Table 1).

Even though the research on thriving at work has advanced rapidly since the high-quality empirical publication in Information System Research in 2001, the dynamic of this good psychological state has not been adequately examined and analyzed statistically and qualitatively [18,35]. Hence, the systematic review justifies gathering the literature on thriving at work through bibliometric analysis and science mapping. These techniques for reviewing are contemplated to determine the broad state and route of any underlying scholarly domain [1,38]. Additionally, the present study used content analysis on high-quality journal publications to identify and categorize research paths, enhancing this literature review's objectivity, breadth, and comprehensiveness. Consequently, this technique aims to disclose the dynamics and trajectory of the research in the management domain using content analysis and quantitative bibliometric review. Therefore, this bibliometric mapping tries to answer the following research question:

RQ: What are the significant trends, broad states, and major trajectories for thriving at work in the management and business domain?

The findings of this systematic review support the researcher and practitioners regarding the research stream on thriving at work in the management domain. Most specifically, this bibliometric mapping review discloses the dynamics and trajectory in the management field that offers a comprehensive overview of academia. Our review acknowledges well-being, health, stress, resources, work engagement, satisfaction, performance, leadership,

motivation, psychological safety, self-efficacy, creativity, and innovation by clustering high-impact articles. Moreover, the evidence of the themes gives an insightful consideration of the relevant study field that may help guide the researchers for further investigation on thriving at work.

The remaining bibliometric mapping review is structured as follows. The second section addresses the deployed method to inquire about the WoS database. In the third section, the broad analysis of thriving at work literature from 2001 to 2021 through WoS has been provided, and later emphasis is resided on mapping the bibliographic data using VOSviewer software (version 1.6, VOSviewer has been developed by Nees Jan van Eck and Ludo Waltman at Leiden University's Centre for Science and Technology Studies (CWTS)). The discussion and conclusion are presented in the last fourth section.

## 2. Methods

To achieve the objective of this research, we conduct a bibliometric analysis using the data from Web of Science's primary collection in the period 2001–2021. We map the emergence and development of the construct thriving at work through this data. Likewise, we identified the mainstream of studies related to this topic through a content analysis of the resultant clusters of most popular keywords and the co-citation analysis by cited references. It is important to say that, for the VOSviewer software, a network is comprised of items and links between them. These items that belong to only one cluster are grouped in clusters that are represented on maps and labeled using numbers.

The analyses included the total link strength attributes of the different items analyzed (i.e., journal citations, documents citations, co-citation, keywords occurrences), understood as a standardized weight attribute. According to the VOSviewer manual, "for a given item, the links and total link strength attributes indicate, respectively, the number of links of an item with other items and the total strength of the links of an item with other items. For example, in the case of co-authorship links between researchers, the links attribute indicates the number of co-authorship links of a given researcher with other researchers. The total link strength attribute indicates the total strength of the co-authorship links of a given researcher with other researchers" (Available online: https://www.vosviewer.com/documentation/Manual_VOSviewer_1.6.8.pdf, accessed on 3 May 2022) The research process involved a set of consecutive stages, which are presented in Figure 1.

*Composition of Bibliometric Data*

Following a consecutive set of procedures, a bibliometric data search was conducted in July 2021. First, we compiled an overview of the advances in the literature on the concept of thriving at work without limit of areas or fields of knowledge from 2001 and 2021 in The Web of Science (WoS) database Core Collection was used. The first search query was: TOPIC: (thriving at work) Timespan from 2001 to 2021. Indexes SCI-EXPANDED, SSCI, A&HCI, BKCI-S, BKCI-SSH, ESCI. As a result, we found 1064 products. Then, we filtered by articles that returned 948 records, which correspond to 89% of all academic products. Excluding the 95 articles that had been produced so far in 2021 (we only compared finished years), a growing interest in the topic of thriving at work is observed. Figure 1 shows how the remaining 853 articles are distributed between 2001 and 2020, showing how this concept grew progressively (Figure 2).

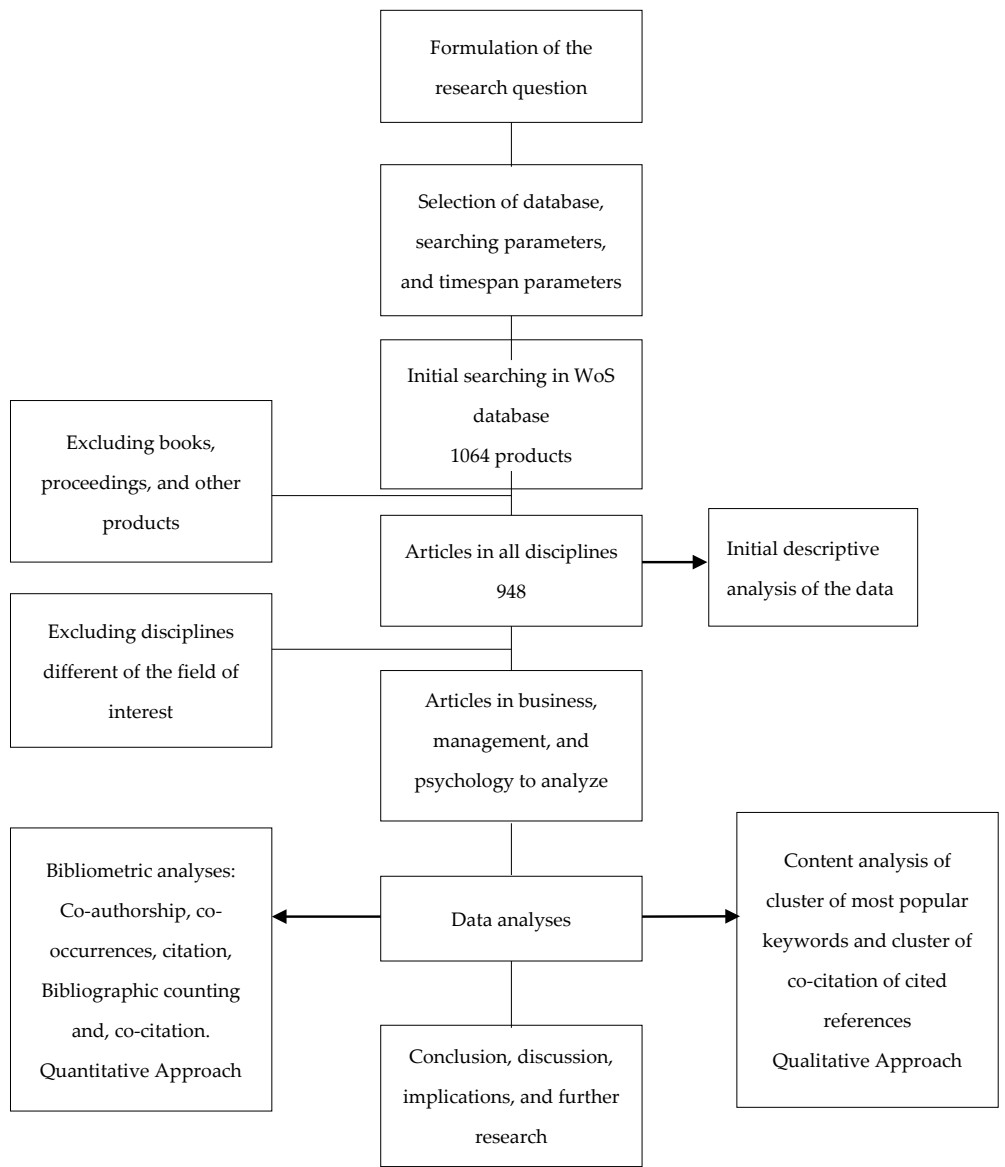

**Figure 1.** Stages of the research process, following the bibliometric mapping research methodology. The arrows are related to the findings. Source: Authors' elaboration.

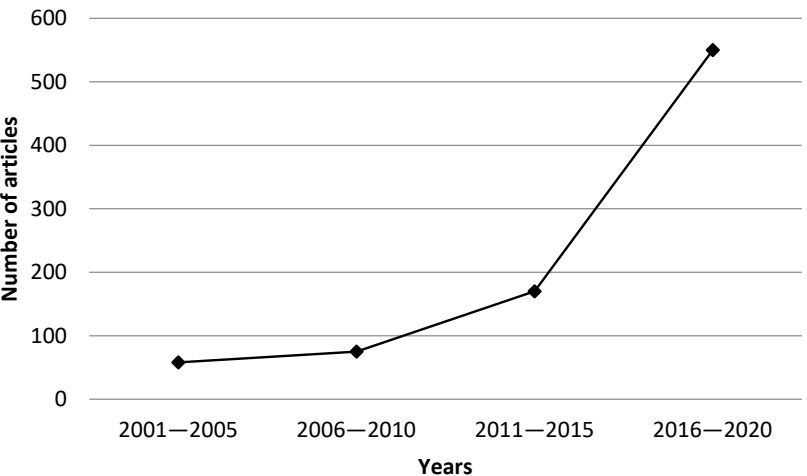

**Figure 2.** Number of publications grouped in periods of five years between 2001 and 2020.

In order to conduct the appropriate bibliometric analysis to pinpoint the said topic in the light of literary efforts made in past, we accounted for the articles which addressed the topic of thriving at work from the time span of 2001 to 2020. The 948 articles were considered to conduct the second search. The search query was: TOPIC: (thriving at work) Timespan from 2001 to 2021. Indexes SCI-EXPANDED, SSCI, A&HCI, BKCI-S, BKCI-SSH, ESCI. Document Types: Articles. Web of Science Categories: Management or Psychology, Applied or Business or Psychology, Multidisciplinary. As a result of this search, we obtained 190 articles. These 190 articles are distributed in different WoS categories. Figure 3 shows the five main literary fields and their resulted percentages from the bibliometric analysis. It can be witnessed that management and business jointly comprised 61% of the pie (Figure 3).

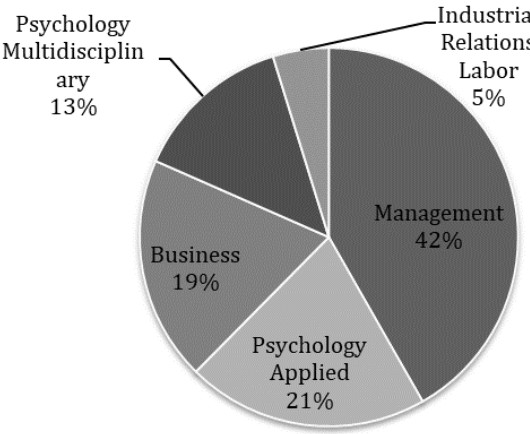

**Figure 3.** Five literary domains of 190 articles.

The countries that have made larger contributions to the notion of thriving at work are: The United States, China, Australia, Pakistan, and England (Figure 4).

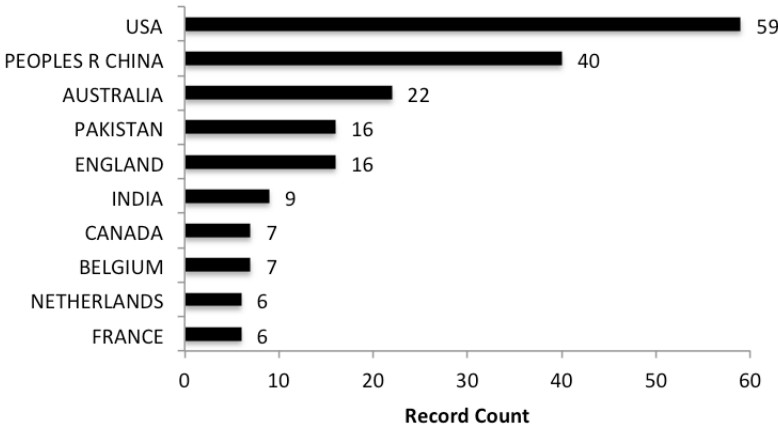

**Figure 4.** The highest contributing countries to the 'thriving at work' concept.

## 3. Results

### 3.1. Bibliometric Analysis

This section presents the main findings of this bibliometric study through the VOSviewer version 1.6. According to researchers [1,38], this software is one of the most widely accepted in academic research.

### 3.1.1. Highly Cited Articles

We selected the articles with at least 10 citations (threshold) to account for the most cited article in the dataset of 190 documents. The article which met the criteria to be more cited was 63. The most-cited article belonged to researchers [39], titled "Explaining the Relationships between Job Characteristics, Burnout, and Engagement: The Role of Essential

Psychological Need Satisfaction", published in *Work & Stress*, with 525 citations. It could be considered a highly influential article.

Table 2 shows the top 10 most-cited articles with their number of citations received.

**Table 2.** The 10 most-cited articles in the topic thriving at work.

| Authors | Year | Title | Journal | Citation | Link Strength |
|---|---|---|---|---|---|
| Van den Broeck et al. [39] | 2008 | "Explaining the Relationships between Job Characteristics, Burnout, and Engagement: The Role of Basic Psychological Need Satisfaction" | *Work & Stress* | 525 | 2 |
| Porath et al. [27] | 2012 | "Thriving at Work: Toward its Measurement, Construct Validation, and Theoretical Refinement" | *Journal of Organizational Behavior* | 238 | 51 |
| Fournier and Lee [40] | 2009 | "Getting Brand Communities Right" | *Harvard business review* | 221 | 0 |
| Paterson et al. [26] | 2014 | "Thriving at Work: Impact of Psychological Capital and Supervisor Support" | *Journal of Organizational Behavior* | 144 | 32 |
| Prilleltensky [41] | 2012 | "Wellness as Fairness" | *American Journal of Community Psychology* | 134 | 1 |
| Wallace et al. [42] | 2016 | "A Multilevel Model of Employee Innovation: Understanding the Effects of Regulatory Focus, Thriving, and Employee Involvement Climate" | *Journal of Management* | 123 | 25 |
| Spreitzer et al. [5] | 2012 | "Toward Human Sustainability: How to Enable More Thriving at Work" | *Organizational Dynamics* | 122 | 30 |
| Niessen et al. [6] | 2012 | "Thriving at Work—A Diary Study" | *Journal of Organizational Behavior* | 103 | 26 |
| Hauge et al. [43] | 2009 | "Individual and Situational Predictors of Workplace Bullying: Why Do Perpetrators Engage in the Bullying of Others?" | *Work & Stress* | 98 | 0 |
| Prem et al. [44] | 2017 | "Thriving on Challenge Stressors? Exploring Time Pressure and Learning Demands as Antecedents of Thriving at Work" | *Journal of Organizational Behavior* | 92 | 16 |

Figure 5 shows how these authors with their articles are related to each other.

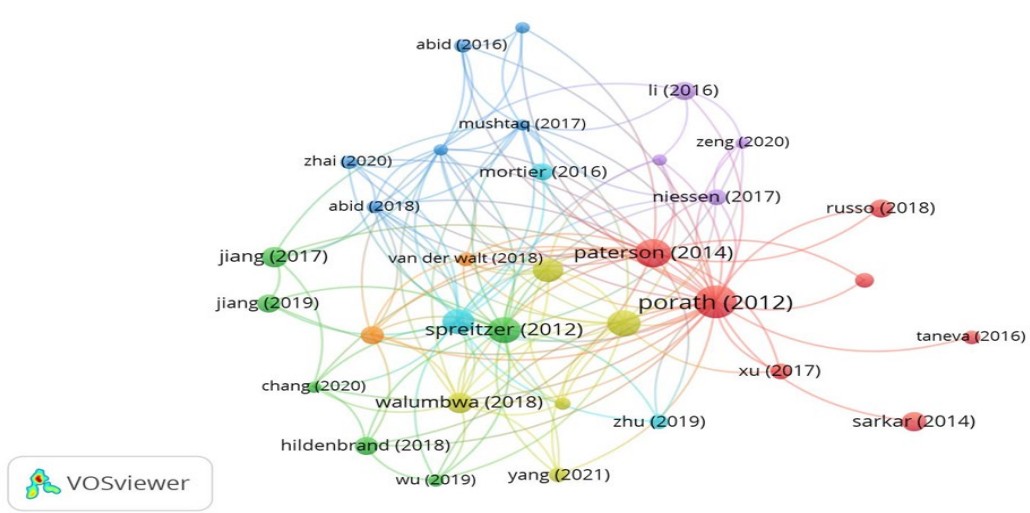

**Figure 5.** Bibliometric map of authors of the most-cited articles and links between them.

### 3.1.2. The Most Productive and Highly Cited Journals

The most cited journal with at least three papers on thriving at work is the Journal of Organizational Behavior, with 674 citations (Table 3). Regarding the number of articles, *Frontiers in Psychology* is the journal with the most articles (13) followed by *Harvard Business Review* (9) and *Journal of Organizational Behavior* (7). *Journal of Organizational Behavior* is the scientific journal with the highest link strength, showing that it is the most cited together with others on the topic of thriving at work. In this journal, there is the largest co-citation network on this topic.

**Table 3.** Highly cited journals in the topic of thriving at work organized by several citations.

| Source | Number of Documents | Number of Citations | Total Link Strength |
|---|---|---|---|
| *Journal of Organizational Behavior* | 7 | 674 | 261 |
| *Work & Stress* | 3 | 641 | 2 |
| *Harvard Business Review* | 9 | 328 | 0 |
| *Journal of Management* | 4 | 166 | 55 |
| *American Journal of Community Psychology* | 2 | 137 | 2 |
| *Organizational Dynamics* | 1 | 122 | 50 |
| *Journal of Vocational Behavior* | 4 | 96 | 49 |
| *Journal of Career Development* | 2 | 88 | 3 |
| *Journal of Nursing Management* | 2 | 72 | 11 |
| *Sport Exercise and Performance Psychology* | 1 | 53 | 2 |

Figure 6 shows how these journals with their articles are related to each other.

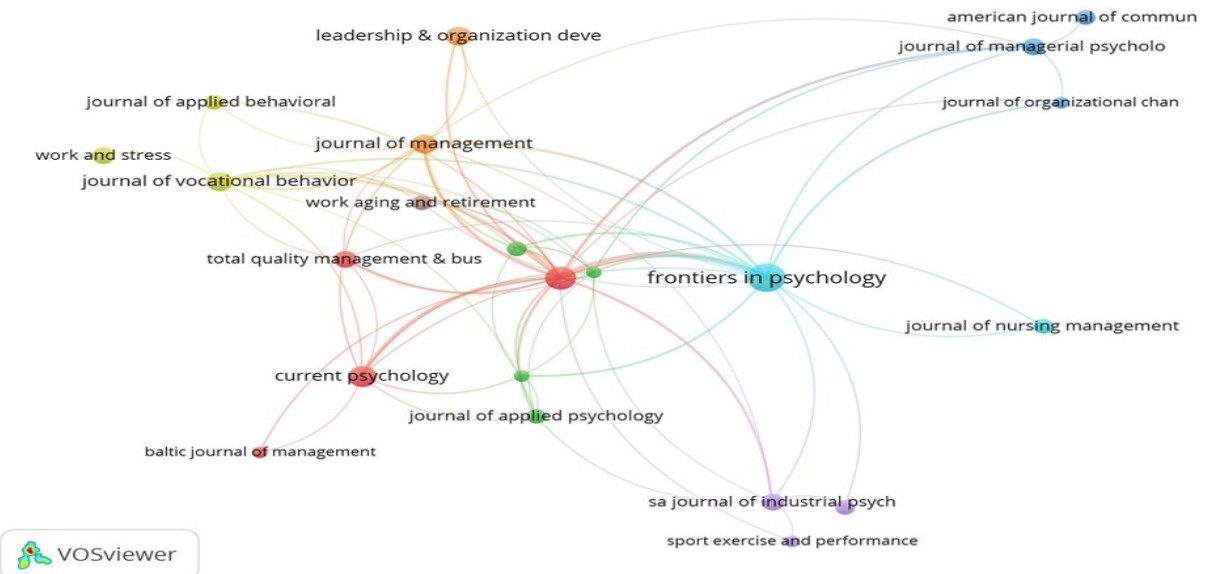

**Figure 6.** The most cited journals and the links between them.

### 3.1.3. The Most Influential Authors

Regarding the authors, we select the highly cited authors in thriving at work. Of the 480 authors, De Witte, Lens, Van den Broeck and Vansteenkiste are the most cited authors with 525 citations. Table 4 presents the top ten more cited authors with their number of citations and linkage with others.

**Table 4.** The 10 most-cited authors in thriving at work, citations linked with others.

| Author | Number of Documents | Number of Citations | Total Link Strength |
|---|---|---|---|
| De Witte, Hans | 1 | 525 | 6 |
| Lens, Willy | 1 | 525 | 6 |
| Van den Broeck, Anja | 1 | 525 | 6 |
| Vansteenkiste, Maarten | 1 | 525 | 6 |
| Spreitzer, Gretchen | 4 | 403 | 547 |
| Garnett, Flannery G. | 1 | 238 | 317 |
| Gibson, Cristina | 1 | 238 | 317 |
| Porath, Christine | 1 | 238 | 317 |
| Fournier, Susan | 1 | 221 | 0 |
| Lee, Lara | 1 | 221 | 0 |

Regarding the number of documents, the following five are the most productive: Abid, G., Jian, Z., Spreitzer, G., Farooqi, S., and Walumbwa, F. (With 8, 5, 4, 4, and 3 documents, respectively).

### 3.1.4. Co-Authorship Analysis between Highly Cited Authors

Regarding co-authorship, we found 480 authors; we selected authors with at least 10 citations. From them, 185 meet the threshold. The total strength of co-authorship links with other authors was calculated. From this analysis, a list of the 15 most cited and connected was derived. The authors with the greatest total link strength are Ghulam Abid and Saira Farooqi (Table 5). It is interesting to note the highest link strength of Abid, Ghulam, and Farooqi, Saira, which is the strongest total link in the authors' network. This indicates that these two authors are often cited together when studying thriving at work.

**Table 5.** Highly cited authors with the greatest total link strength.

| Author | Number of Documents | Number of Citations | Total Link Strength |
| --- | --- | --- | --- |
| Abid, Ghulam | 8 | 80 | 19 |
| Farooqi, Saira | 4 | 43 | 13 |
| Hoefer, Stefan | 3 | 15 | 11 |
| Hoege, Thomas | 3 | 15 | 11 |
| Huber, Alexandra | 3 | 15 | 11 |
| Strecker, Cornelia | 3 | 15 | 11 |
| Arya, Bindu | 3 | 30 | 9 |
| Sreitzer, Gretchen | 4 | 403 | 9 |
| Elahi, Natasha Saman | 2 | 25 | 7 |
| Walumbwa, Fred O. | 3 | 65 | 7 |
| Misati, Everlyne | 2 | 63 | 6 |
| Porath, Christine l. | 2 | 163 | 6 |
| Ahmed, Samiah | 2 | 21 | 5 |
| Brenner, Mirja | 1 | 14 | 5 |
| Hausler, Melanie | 1 | 14 | 5 |

Figure 7 shows comparative strengths and interlinks among the authors.

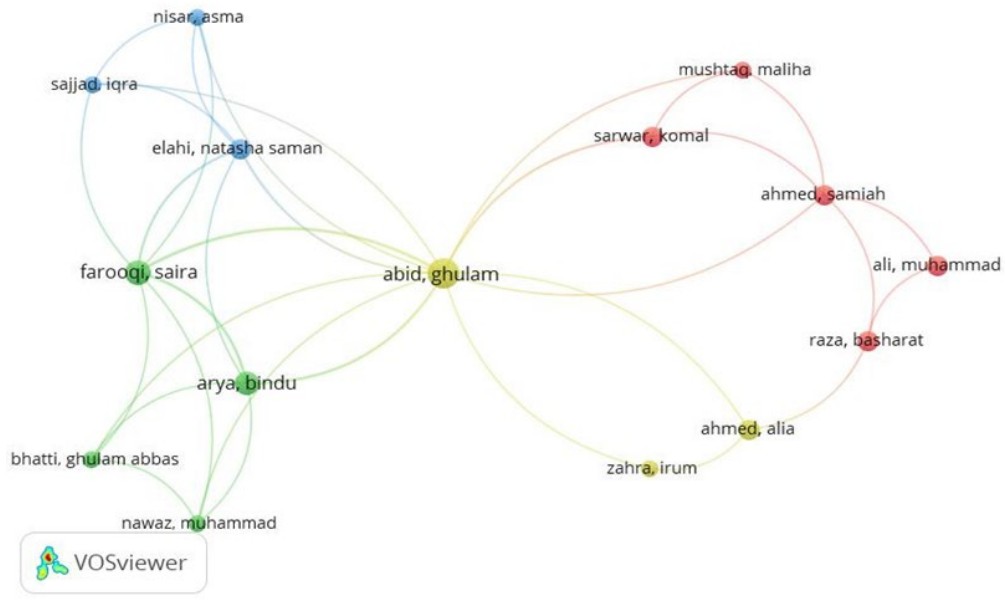

**Figure 7.** The total strength of co-authorship links between the 15 authors is highly connected with other authors.

### 3.1.5. Co-Authorship Analysis Considering the Countries as the Unit of Analysis

This analysis was conducted by selecting the countries with at least five articles. Of the 36 countries, 14 meet the threshold. Excluding South Africa with papers but not links with others, we analyze 13 countries. These countries are grouped in four clusters where they are highly connected. Cluster 1: Pakistan, Peoples Republic of China, South Korea, USA; Cluster 2: Belgium, Canada, France, Netherlands; Cluster 3: Australia, England, India; and Cluster 4: Austria, Germany (Figure 8). As can be seen, excepting Pakistan and India, thriving at work has been more studied in developed countries.

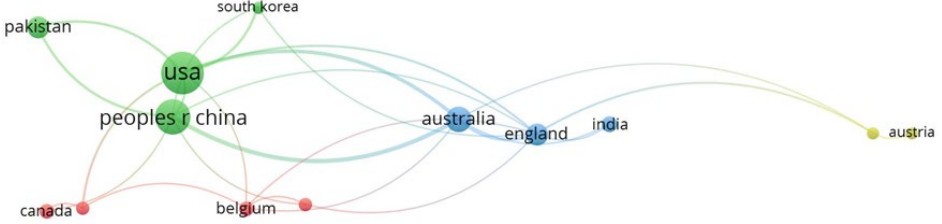

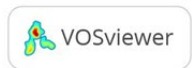

**Figure 8.** Co-authorship networks between countries with more contribution and cooperation between them.

### 3.1.6. Co-Citation Analysis of Cited Sources

Co-citation analysis of cited sources was also conducted. We select 50 as the minimum number of citations of a source. We decided to take the threshold of 50 since the 10 most cited journals are just above this citation level. Of the 3402 sources, 32 meet the threshold. The sources with the greatest total link strength were selected. Table 6 shows the 10 sources with more citations and more correlation between sources. In this regard, Table 6 shows the more influential sources in thriving at work.

**Table 6.** Ten more influential journals in the topic of thriving at work in terms of co-citation and relation between them.

| Journal | Citations | Link Strength |
|---|---|---|
| *Academy of Management Journal* | 419 | 16,742 |
| *Academy of Management Review* | 243 | 9082 |
| *Administrative Science Quarterly* | 120 | 4754 |
| *American Psychologist* | 151 | 4650 |
| *Annual Review of Psychology* | 98 | 3719 |
| *European Journal of Work and Organizational Psychology* | 53 | 1849 |
| *Frontiers in Psychology* | 67 | 2121 |
| *Harvard Business Review* | 50 | 1931 |
| *Human Relations* | 89 | 3253 |
| *Human Resource Management* | 62 | 2446 |

It can be seen that the 10 most influential journals in thriving at work are distributed in similar proportion in administrative sciences (5), psychology (4), and one where these two areas of knowledge are highly involved—Figure 9 maps the linkages between these sources.

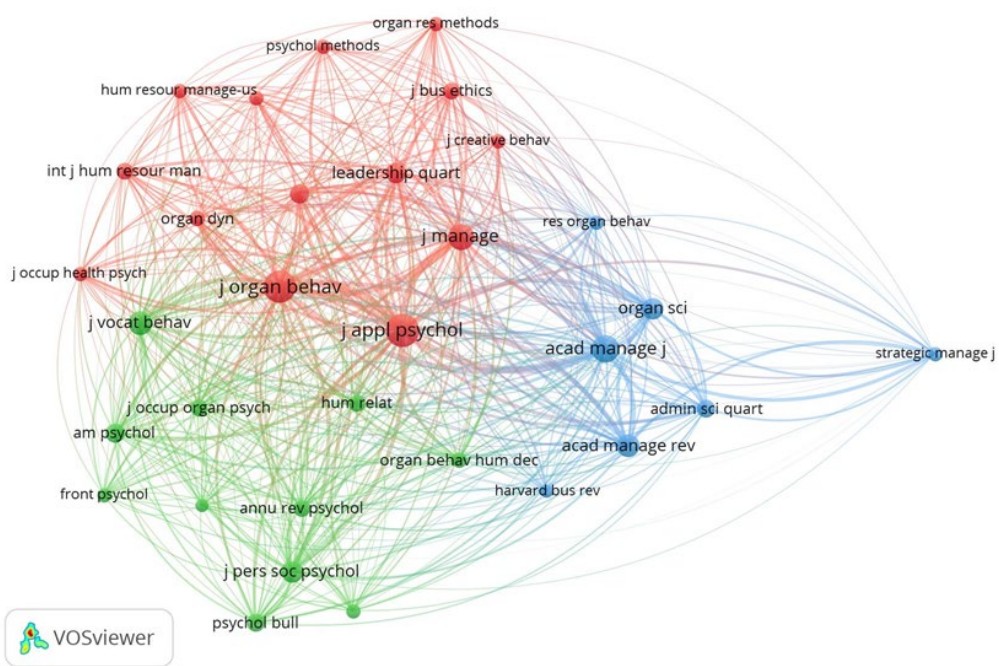

**Figure 9.** Map of co-citation of cited journals.

### 3.1.7. Co-Citation Analysis of Cited Authors

We conduct a co-citation analysis of authors to identify the networks among them. Of the 6515 authors, we select authors with a minimum of 50 citations; 15 meet the threshold (Table 7). Figure 10 allows identifying how this network is organized. Spreitzer, G. Porath, C., and Podsakoff, P. are the most influential authors in this identified network.

### 3.1.8. Bibliographic Coupling by Documents

The bibliographic coupling allows identifying the papers that use the same set of cited articles and shows the authors bibliographically coupled by clusters showing the most influential authors and identifying the networks that the authors are built between them (Table 8).

**Table 7.** Co-citation of cited authors and total link strength.

| Authors | Citations | Total Link Strength |
|---|---|---|
| Spreitzer, G. | 205 | 1822 |
| Porath, C. | 106 | 983 |
| Podsakoff, P. | 90 | 800 |
| Carmeli, A | 83 | 887 |
| Ryan, R. | 72 | 699 |
| Hobfoll, S. | 64 | 558 |
| Niessen, C. | 62 | 700 |
| Paterson, T. | 60 | 636 |
| Spreitzer, G. | 59 | 686 |
| Abid, G. | 55 | 595 |
| Fredrickson B. | 55 | 470 |
| Bandura, A. | 52 | 505 |
| Deci, E. | 52 | 521 |
| Preacher, K. | 52 | 515 |
| Bakker, A. | 51 | 485 |

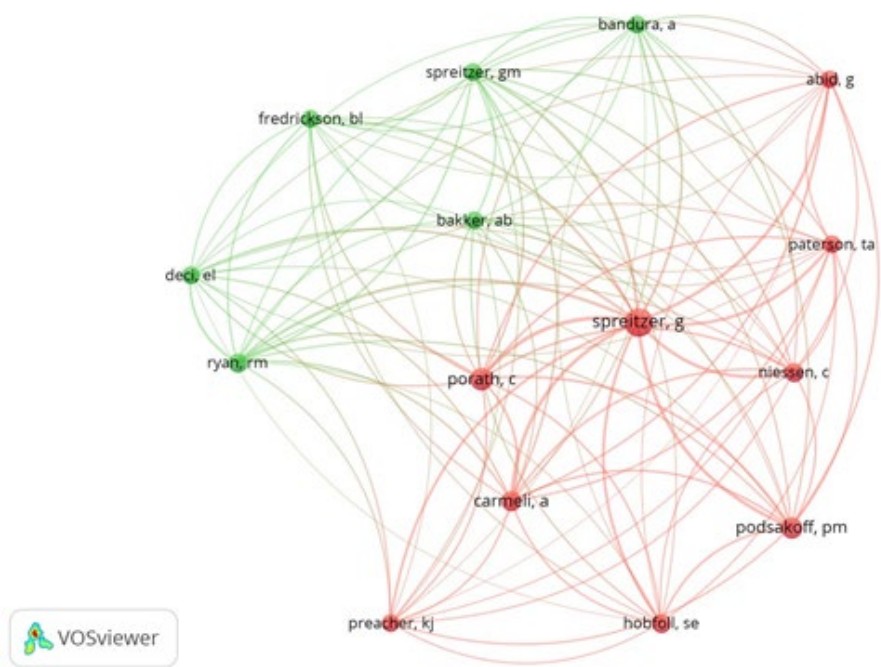

**Figure 10.** Map of co-citation of cited authors that show how the networks are grouped.

**Table 8.** Bibliographic coupling of articles.

| Authors | Article | Citations | Link Strength |
|---|---|---|---|
| Wallace et al. [42] | "A Multilevel Model of Employee Innovation: Understanding the Effects of Regulatory Focus, Thriving, and Employee Involvement Climate" | 123 | 126 |
| Niessen et al. [6] | "Thriving at Work—A Diary Study" | 103 | 125 |
| Paterson et al. [26] | "Thriving at Work: Impact of Psychological Capital and Supervisor Support" | 144 | 119 |
| Niessen et al. [19] | "Thriving when Exhausted: The Role of Perceived Transformational Leadership" | 30 | 115 |
| Prem et al. [44] | "Thriving on Challenge Stressors? Exploring Time Pressure and Learning Demands as Antecedents of Thriving at Work" | 92 | 107 |
| Jiang et al. [14] | "Knowledge Hiding as a Barrier to Thriving: The Mediating Role of Psychological Safety and Moderating Role of Organizational Cynicism" | 39 | 104 |
| Walumbwa et al. [18] | "Inspired to Perform: A Multilevel Investigation of Antecedents and Consequences of Thriving at Work" | 56 | 102 |
| Gerbasi et al. [45] | "Destructive De-Energizing Relationships: How Thriving Buffers their Effect on Performance" | 41 | 98 |
| Porath et al. [27] | "Thriving at Work: Toward Its Measurement, Construct Validation, and Theoretical Refinement" | 238 | 96 |
| Yang et al. [46] | "Why and When Paradoxical Leader Behavior Impact Employee Creativity: Thriving at Work and Psychological Safety" | 25 | 94 |

3.1.9. Bibliographic Coupling of Journals

For this analysis, we select sources with a minimum of two documents; of the 114 sources found, 35 meet the thresholds. Table 9 presents the 10 top journals.

**Table 9.** Bibliographic coupling of journals.

| Journal | Documents | Citations | Link Strength |
|---|---|---|---|
| *Frontiers in Psychology* | 13 | 43 | 3196 |
| *Journal of Organizational Behavior* | 7 | 674 | 2386 |
| *Current Psychology* | 5 | 28 | 1363 |
| *Journal of Vocational Behavior* | 4 | 96 | 1269 |
| *Personnel Review* | 3 | 2 | 1188 |
| *Journal of Management* | 4 | 166 | 1150 |
| *International Journal of Contemporary Hospitality Management* | 3 | 19 | 1012 |
| *Journal of Managerial Psychology* | 3 | 21 | 814 |
| *Human Resource Management* | 2 | 30 | 810 |
| *European Journal of Training and Development* | 2 | 0 | 709 |

3.1.10. Bibliographic Coupling of Authors

This analysis shows the authors that are bibliographically coupled. We select authors with a minimum of two articles and at least 10 citations. Of the 480 authors, 36 meet the threshold. For each of the 36 authors, the total strength of the bibliographic coupling links with other authors is calculated. The author most represented is Spreitzer, Gretchen. The top 10 most influential authors considering the bibliographic coupling are presented in Table 10.

**Table 10.** The 10 most representative authors by bibliographic coupling analysis.

| Authors | Documents | Citations | Total Link Strength |
|---|---|---|---|
| Spreitzer, Gretchen | 4 | 403 | 1230 |
| Porath, Christine L. | 2 | 163 | 611 |
| Niessen, Cornelia | 2 | 133 | 922 |
| Gibson, Cristina B. | 2 | 124 | 307 |
| Jiang, Zhou | 5 | 106 | 2058 |
| Abid, Ghulam | 8 | 80 | 5197 |
| Walumbwa, Fred O. | 3 | 65 | 1810 |
| Misati, Everlyne | 2 | 63 | 1186 |
| Muchiri, Michael K | 2 | 59 | 1292 |
| Jiang, Xuan | 2 | 45 | 1191 |

Figure 11 shows the networks created between the authors according to bibliometric coupling analysis.

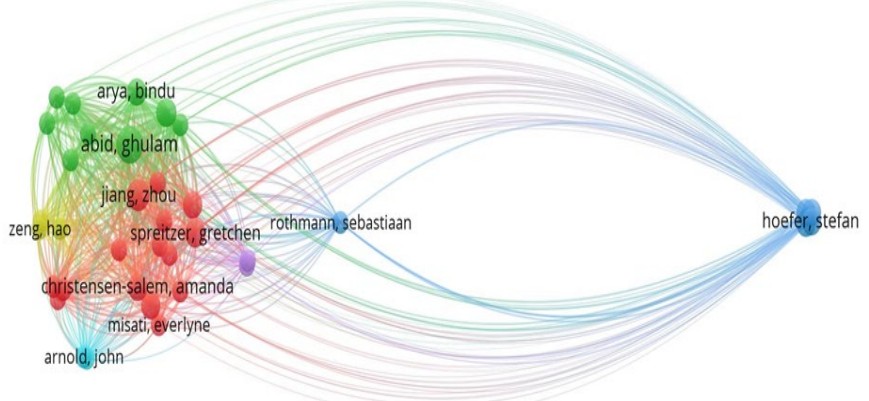

**Figure 11.** Bibliographic coupling of authors.

3.1.11. Co-Occurrence Analysis for Popular Keywords

This analysis is conducted to observe the scope of the study of thriving at work research. The most popular keywords were in total 993. We selected a minimum number

of keyword occurrences of at least 10 times, for which 32 met the threshold. From each of the 32 keywords, the total strength of the co-occurrence links with other keywords was calculated. The 10 keywords with high total link strengths are shown in Table 11.

**Table 11.** The most popular keywords in the topic of thriving at work.

| Keyword | Number of Occurrences | Total Link Strength |
|---|---|---|
| Thriving at work | 48 | 183 |
| Performance | 46 | 155 |
| Model | 31 | 119 |
| Work | 29 | 110 |
| Thriving | 29 | 101 |
| Motivation | 23 | 91 |
| Antecedents | 22 | 95 |
| Impact | 21 | 91 |
| Mediating role | 20 | 94 |
| Transformational leadership | 19 | 78 |

Supported by these most popular keywords, it seems clear that the study of thriving at work has been focused on creating models to estimate its impact on performance, where the most frequently included variables are related to antecedents, motivation, and transformational leadership.

*3.2. Content Analysis*

3.2.1. Content Analysis of the Most Popular Keywords

Content analysis was conducted with the resultant clusters of the most popular keywords. These popular keywords, grouped in clusters, represent the main research streams in thriving at work in the business management and psychology field. Cluster 1 comprises 11 keywords: work, resources, and engagement, which were the most popular with 29, 18, and 16. Cluster 2 comprises 11 keywords. Thriving, motivation, and mediating roles are the most popular with 29, 23, and 20. Cluster 3 comprises 10 keywords. Thriving at work, performance, and model were the most popular with 48, 46 and 31 occurrences, respectively. The content analysis of the most popular keywords grouped in the three clusters; represents the main research themes in thriving at work (Table 12).

In Figure 12, the network of links between the most popular keywords can be seen.

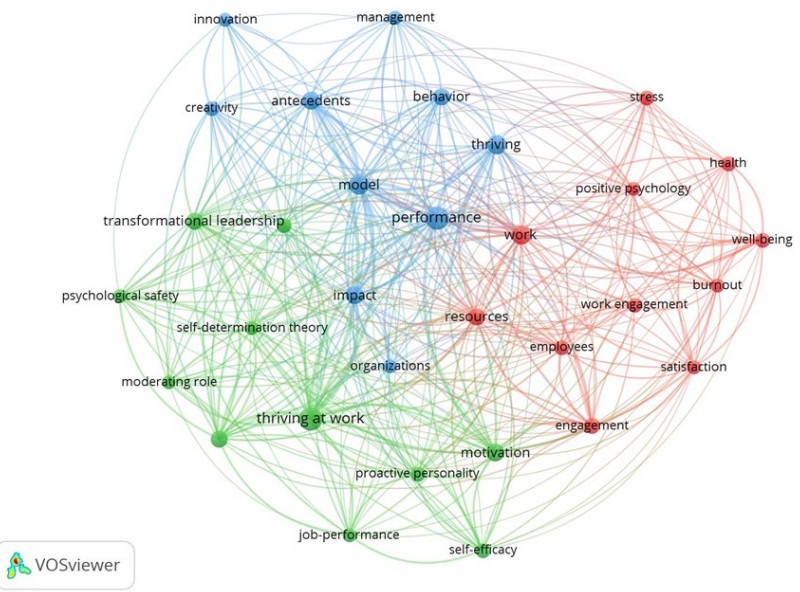

**Figure 12.** Mapping links between the most popular keywords conform to three clusters.

**Table 12.** Content analysis of the three resultant clusters of most popular keywords in the topic of thriving at work.

| | Keyword | Occurrences | Total Link Strength | Content Analysis |
|---|---|---|---|---|
| Cluster 1 (11 documents) | Work | 29 | 110 | Employees' personal experience at work related to their psychological state |
| | Resources | 18 | 70 | |
| | Engagement | 16 | 66 | |
| | Well-being | 13 | 42 | |
| | Employees | 12 | 47 | |
| | Health | 12 | 35 | |
| | Stress | 12 | 33 | |
| | Work engagement | 12 | 33 | |
| | Burnout | 11 | 43 | |
| | Satisfaction | 11 | 44 | |
| | Positive psychology | 10 | 36 | |
| Cluster 2 (11 documents) | Thriving | 29 | 101 | Mediating and moderating models of thriving at work, framed mainly in the self-determination theory |
| | Motivation | 23 | 91 | |
| | Mediating role | 20 | 94 | |
| | Transformational leadership | 19 | 78 | |
| | Leadership | 14 | 41 | |
| | Proactive personality | 14 | 57 | |
| | Self-efficacy | 12 | 54 | |
| | Job-performance | 11 | 46 | |
| | Psychological safety | 11 | 54 | |
| | Moderating role | 10 | 43 | |
| | Self-determination theory | 10 | 53 | |
| Cluster 3 (10 documents) | Thriving at work | 48 | 183 | Organizational outcomes, mainly in terms of performance, creativity, and innovation related to thriving at work |
| | Performance | 46 | 155 | |
| | Model | 31 | 119 | |
| | Antecedents | 22 | 95 | |
| | Impact | 21 | 91 | |
| | Behavior | 19 | 73 | |
| | Creativity | 12 | 49 | |
| | Management | 11 | 40 | |
| | Organizations | 11 | 43 | |
| | Innovation | 10 | 25 | |

### 3.2.2. Content Analysis of the Co-Citation of Cited References

This analysis was conducted to identify the main orientations of the study of thriving at work. The content analysis of 190 articles was conducted using the data clustering technique. The co-citations determine the correlation between the publications in terms of content, in other words, in the co-citation analysis, it is assumed that publications are cited together frequently because these are similar thematically.

The 190 articles have 9642 cited references; from them, we select the minimum number of citations of cited references 20, obtaining 23 that meet the threshold. We decided to take the threshold of 20 since the higher cited references are above this number. These 23 cited references were grouped in three clusters. The analysis of each cluster allowed for the identification of the main themes and the research approach used to study this topic. The three resultant clusters that show the bibliometric map of cited references are presented in Table 13.

**Table 13.** Content analysis of the three clusters was obtained through the co-citation analysis by cited references.

| Authors | Topic | Methodological Parameters | Citations | Link Strength |
|---|---|---|---|---|
| Paterson et al. [26] | The study tests the relationship between thriving at work and self-development and agentic work behaviors. Psychological capital and supervisor support climate as antecedent variables are proposed and tested. | Type of research: Quantitative Type of article: Empirical study | 60 | 496 |
| Carmeli and Spreitzer [47] | Trust, connectivity, and thriving as drive employees' innovative behaviors in the work context are analyzed. | Type of research: Quantitative Type of article: Empirical study | 50 | 444 |
| Spreitzer et al. [5] | Thriving at work is defined, and what can be done to enhance thriving at work is discussed to contribute to the sustainability. | Type of research: Quantitative Type of article: Theoretical/conceptual study | 50 | 433 |
| Niessen et al. [6] | Employees thriving at work in response to resources such as positive meaning, relational resources, and knowledge is studied. | Type of research: Quantitative Type of article: Empirical study | 47 | 427 |
| Wallace et al. [42] | It is examined the effects of thriving on employee involvement climate, and employee regulatory focus to innovation through a multilevel model. | Type of research: Quantitative Type of article: Empirical study | 34 | 328 |
| Walumbwa et al. [18] | The contextual and individual factors that facilitate thriving at work are examined through a multilevel model. The relationship with health and performance is analyzed. | Type of research: Quantitative Type of article: Empirical study | 32 | 318 |
| Prem et al. [44] | The effect of time pressure and learning demands, as stressors on thriving at work, is examined. | Type of research: Quantitative Type of article: Empirical study | 28 | 260 |
| Spreitzer and Porath [48] | Thriving at work is discussed, and it is proposed an integrative model of human growth in the work context. | Type of research: Quantitative Type of article: Theoretical/conceptual study | 23 | 225 |
| Jiang [9] | The relationship between proactive personality and career adaptability is analyzed through a moderated and mediation approach. | Type of research: Quantitative Type of article: Empirical study | 21 | 211 |
| Hu and Bentler [49] | Conventional cut-off criteria and the new statistical alternatives for fit indexes used to evaluate models are discussed. | Type of research: Quantitative Type of Article: Theoretical/conceptual study | 22 | 189 |
| Cluster 1 (10 documents) Label: Identification of variables related to thriving at work | | | | |
| Spreitzer et al. [3] | A model of thriving at work as a theoretical construct, compared with similar notions is developed. This study makes a theoretical contribution. | Type of research: Qualitative Type of article: Theoretical/conceptual study | 100 | 707 |
| Porath et al. [27] | A validated measurement of the construct of thriving at work is developed. | Type of research: Quantitative Type of article: Empirical study | 96 | 704 |
| Nix et al. [50] | The effects of experimentally induced motivational orientations on the positive affect of vitality and happiness are examined. | Type of research: Quantitative Type of article: Empirical study | 27 | 230 |

**Table 13.** *Cont.*

| Authors | Topic | Methodological Parameters | Citations | Link Strength |
|---|---|---|---|---|
| Kleine [51] | Thriving is analyzed in relation to psychological capital, proactive personality, positive affect, work engagement, support perceived, performance, health, burnout, commitment, and job satisfaction. | Type of research: Quantitative Type of article: Empirical study (Metha-analysis) | 23 | 207 |
| Carver [52] | The distinction between resilience and thriving is addressed. | Type of research: Quantitative Type of article: Theoretical/conceptual study | 23 | 193 |
| Aiken [53] | The structure, test, and interpretation of multiple regression models are discussed. | Type of research: Quantitative Type of article: Theoretical/conceptual study | 20 | 153 |
| Hobfoll [54] | An alternative model of stress supported in the model of conservation of resources is presented. | Type of research: Quantitative Type of article: Theoretical/conceptual study | 20 | 124 |
| Cluster 2 (7 documents) Label: Concept delimitation and accuracy of its measurement | | | | |
| Deci and Ryan [55] | Growth, integrity, and human well-being are analyzed, framed in self-determination theory (SDT). | Type of research: Quantitative Type of article: Theoretical/conceptual study | 23 | 174 |
| Fredrickson [56] | A new theoretical perspective on positive emotions in the perspective of positive psychology is described. | Type of research: Quantitative Type of article: Theoretical/conceptual study | 28 | 179 |
| Ryan and Deci [57] | Self-motivation, healthy psychological development, and psychological needs are studied under self-determination theory (SDT). | Type of research: Quantitative Type of article: Theoretical/conceptual study | 23 | 171 |
| Cluster 3 (3 documents) Label: Concept delimitation and accuracy of its measurement | | | | |

*3.3. Content Analysis of Clusters*

3.3.1. Cluster 1: Identification of Variables Related to Thriving at Work

This cluster tests different individual and contextual variables that can enhance thriving at work in the employees. Most of the studies in this domain are quantitative and empirical in nature (8 studies out of 10).

3.3.2. Cluster 2: Concept Delimitation and Accuracy of Its Measurement

Research studies in this cluster revolve around the concept of thriving at work through conceptual discussions, meta-analyses, and differentiation with similar constructs. Likewise, this cluster intends to obtain accurate scales for measuring the same concept and further discuss the statistical validation of models. Unlike Cluster 1, Cluster 2 is comprised mainly of conceptual and theoretical studies (six studies out of seven) which analyzed the concept and conducted its measurement, including statistical discussion about the model's development.

3.3.3. Cluster 3: Theoretical Support of The Concept

The studies that include this cluster are all theoretical/conceptual (three papers). This cluster addresses the notion of well-being broadly, supported mainly by the self-determination theory (SDT).

**4. Discussion and Future Directions**

The present study aims to address the call from academicians [18,35] to examine and analyze the major trajectories and trends related to thriving at work. Hence, we used bibliometric mapping and content analysis on high-quality publications to identify and categorize research paths. It is proven in this research that studies linked to thriving at work in business and management have extended significantly over the globe over the

previous two decades. From 2012 onward, the trend accelerated, with the most prolific years becoming the most productive (see Figure 2). As some have suggested, the increased number of high-impact publications is not restricted to increasing the number of articles published in this study field by a single journal or by a single nation. Contributions with a high impact have been received worldwide on this subject. Furthermore, it should be noted that the United States of America has provided the greatest number of publications (59), followed by China, Australia, Pakistan, and the United Kingdom (Figure 4). According to the data, the management domain accounted for most articles (42%). Furthermore, it can be noticed that the management and business together achieved 61 percent of the pie. This finding shows that majority of the work on thriving is published in the business domain.

Scholars from the United States, Australia, China, Australia, Pakistan, and England contributed considerably to the thriving at work phenomenon. Ghulam abid, Kinnaird College for Women, has produced eight articles on the said subject with the most substantial link strength and Farooqi, Saira. It is noteworthy to see that Abid, Ghulam, and Farooqi, Saira have the strongest overall link strength in the writers' network. This suggests that these two writers are frequently cited together in studies of workplace thriving. Furthermore, Spreitzer, G. is the most referenced and influential author in this identified network.

Moreover, several top-ranked journals have contributed to the advancement of this discipline. *Harvard Business Review* is the most productive with the output of nine articles, followed by the *Journal of Organizational Behavior* with seven publications. It is also noticed that the bulk of highly referenced publications is published in the *Journal of Organizational Behavior*, with a total of 674 citations belonging to seven works (see Table 3). Furthermore, the *Academy of Management Publication* is the most influential journal of co-citations (see Table 6).

The results also suggested that more useful articles were published by more than two writers, predicting the growing inclination to join up for creating and publishing papers. The findings connected to the high impact citation, publications, journals, and influential authors on thriving at work phenomena in the field of business and management give a chance to acknowledge the improvement of the domain and assess the efforts of multiple players.

The present bibliometric mapping study also delivers significant insight into the evolution and development of thriving at work research in business and management. In this situation, a graphical depiction (Figure 3) displays the most common terms supplied by the writers. This network diagram (see Table 12) has been able to demonstrate essential themes such as engagement, well-being, health, satisfaction, motivation, leadership, performance, personality, creativity, and innovation. These topics involve sub-areas such as innovation, satisfaction, behavior, health, leadership, etc. Hence, this bibliometric study gives useful facts linked to research trends concerning the history, present, and future of working research. The findings give a valuable guide to the researchers who desire to further study this field by conducting research.

The findings suggest that thriving at work research in business and management is still increasing and has its roots in management, entrepreneurship, organizational behavior, leadership, and personality. The current issue is still of interest to academicians because of its good behavioral results. It may be anticipated that academics wanting to research this sector would have an exciting road ahead as there is too much to explore.

*Suggestions for Future Studies*

Based on our content and bibliometric evaluation outcomes, three recommendations for additional research on thriving at work in the sphere of business and management are indicated by clusters. Based on the outcomes of the bibliometric study, the following recommended topics are revealed, i.e., varied research techniques, multidisciplinary studies, and diversified contextual research in the business and management sector.

This bibliometric study of thriving at work mainstream quality publications under consideration incorporates a theoretical approach and quantitative. The literature review,

qualitative research, surveys, and conceptual investigations are employed. Only a few studies have embraced the empirical survey technique designs. We proposed that researchers apply diverse methodologies, especially empirical surveys using multi-time and multi-source data to decrease the prevalent approach biases. Furthermore, future research might highlight the mixed-method design, case studies, and experimental design, suitable for studying the causality.

According to our bibliometric investigation of disciplinary distribution and progression of this phenomenon, research in the business and management field is highly focused on the topics such as engagement, well-being, health, satisfaction, motivation, leadership, performance, personality, creativity, and innovation. It lacks a multidisciplinary influence on other subjects. Likewise, the management and business area has impacted other subjects throughout time, but the number of multidisciplinary studies is considerably small. The majority of the literature is comparable to the business and management sector, such as entrepreneurship, leadership, firm performance, innovation, supply chain management, and performance management. Future studies might cover other multidisciplinary studies, such as operation management, sociology, psychology, and hotel management.

According to the analysis findings, the geographic region of thriving at work literature is generally concentrated more in developed nations, including the United States, China, Australia, Belgium, and France. Future studies might incorporate additional regional studies from emerging nations to increase the feasibility, depth, and breadth of thriving in business and management research.

## 5. Conclusions

Research in business and management published between 2001 and 2021, the WoS database was used to compile a bibliometric analysis of literature on thriving at work. The investigations helped us to track, through 190 papers, the genesis of the notion of thriving at work and its growth. The primary study variables connected to this subject and its approach were discovered. Likewise, the most prominent writers, the most referenced papers, the highly cited journals, and the nations contributing to establishing this concept are found. In addition, an examination of co-citation, co-occurrences, and bibliographic coupling was done. Finally, content analysis of the most frequent keywords and the co-citation of referenced references are undertaken. These studies enable finding the significant developments in the thriving at work phenomenon. Our review acknowledges well-being, health, stress, resources, work engagement, satisfaction, performance, leadership, motivation, psychological safety, self-efficacy, creativity, and innovation by clustering high-impact articles. Moreover, the evidence of the themes provides insightful consideration of the relevant study field that may help guide researchers in further investigation on thriving at work.

**Author Contributions:** G.A. and F.C. contributed equally. All authors have read and agreed to the published version of the manuscript.

**Funding:** This research received no external funding.

**Institutional Review Board Statement:** Not applicable.

**Informed Consent Statement:** Not applicable.

**Data Availability Statement:** Not applicable.

**Acknowledgments:** We thank the Universidad del Rosario, Bogotá-Colombia for the financial support for the publication of this article.

**Conflicts of Interest:** The authors declare that they have no conflict of interest to report regarding the present study.

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
