# Peer review of "Mapping Thriving at Work as a Growing Concept: Review and Directions for Future Studies"

_information, doi:10.3390/info13080383_

Round 1
Reviewer 1 Report
I am happy to review this article. The article is well written. But some suggestions are just for improvement;
My question is to the authors; either they are doing bibliometric analysis after a systematic review or scoping review. Please clear and explain accordingly in methodology.
Please do not use abbreviations in the abstract. In the whole document, the author should first use full text with abbreviations, afterward, he/she can use abbreviations. Some spelling mistakes were found e.g., in figure 3 ta.
Author Response
I have reviewed and accepted the Associate Editor’s Decision, as noted below.
Response to 1st Reviewer’s Comments
Reviewer’s Comment # 1: My question is to the authors; either they are doing bibliometric analysis after a systematic review or scoping review. Please clear and explain accordingly in methodology.
Authors’ Response to Comment # 1: The current study is a bibliometric analysis. We have further improved the manuscript for more clarity.
Reviewer’s Comment # 2: Please do not use abbreviations in the abstract. In the whole document, the author should first use full text with abbreviations, afterward, he/she can use abbreviations. Some spelling mistakes were found e.g., in figure 3 ta.
Authors’ Response to Comment # 2: Thank you for the correction. The linguistics error and other technical flaws are removed through editing the manuscript.
We want to thank the reviewer for the detailed comments that greatly helped us in the revision process. We appreciate the valuable input provided by reviewers and sincerely hope that the revised manuscript adequately addresses all the issues raised.
Reviewer 2 Report
I found it very interesting to read this paper. As I am not an expert in the field of "thriving at work", it is difficult for me to judge the correctness of information about the specific topic of the bibliometric analysis and the literature presented in this article.
The authors have analysed approximately 1000 articles about thriving at work which appeared during a time span of approximately 20 years. The subsequent analysis provides detailed insights into metrics, topics and networks of this field of research. Furthermore, it provides a comprehensive overview over studies and their contents which could be a valuable reference for researchers in that field.
However, this article must be improved considerably in quality which will be detailed in my comments as follows:
ABSTRACT
The abstract is concise and informative.
INTRODUCTION
The introduction is well-written, informative and cites many important articles. However, lines 44-66 are a mere listing without context and explanations. Context and links should be given and the paragraph might be shortened to give the most relevant articles.
67: Not precise enough, what is meant exactly by "outcomes"?
79: This is a strong statement which should have citations supporting it.
95: "acknowledged" -> acknowledges?
98: "the security of these themes" <- what is meant here?
101: "The reset of the bibliometric mapping review is structured as follows." <- rest? remainder?
101-103: no verb
105: were -> are
The introduction is well-structured.
METHODS
110: "mainstreams of study" -> studies?
112: "presented in the figure 1"
117: Caption should be expanded to describe the contents of the figure. Figure quality needs improvement (e.g. clipped text). Is the figure based on a (citable) standard procedure for bibliometric analyses? What do boxes mean, what is the difference between lines and arrows?
131 Figure 2: Quality needs to be improved, figure seems squeezed. Caption needs a detailed description. Tick marks don't match data points. Why is the temporal resolution so low (5 years)?
133: Past and Present tense are mixed and should be standardized, e.g. "we conduct our second searching. Our search query was:" This should be unified throughout the whole article.
138: "Figure 2 shows the five main areas and their percentage. As can be seen, manage-
ment and business joined to achieve the 61% (Figure 3)." -> Fig. 3 instead of Fig. 2, I dont' understand the second sentence. Why were they joined?
142: Caption needs to be corrected and expanded.
144-146: Grammer and language needs to be streamlined. "more than what?" Information about the total number of countries is missing. The plot should be updated: A maximum value of 60 is sufficient for the x axis. Include a tick with a label between zero and 20. It is difficult to guess the numbers for the countries France to India.
RESULTS
156: Split sentence or correct grammar.
157: Using quotation marks or different font style would help to distinguish article titles from article text.
164/171: Figure captions and table captions should always be on the same page as the respective figures.
171: Why is it table 3, where is table 2? Caption not detailed enough. Table formatting should be improved (line breaks etc.). Please explain "link strength" in methods section.
163: Figure 5 might contain very interesting information, but this figure is not interpretable in this form: graph layout needs to be improved for readability, fonts are too small, resolution of figure is not good enough. Figure caption needs to be expanded. Same for: Fig. 6-12. Why is everything in the graph figures in lower case? Please unify this to match the text.
Generally, the text flow in this section is often interrupted as figures, tables and text are entangled to heavily. Please restructure to have a more coherent look and feel of this section.
Also the table formatting needs to be improved for all tables.
188: "The total strength of co-authorship links with other authors was calculated." Please explain the concept of the "total strength of ... links" in the method section. Also, please add a few words about the "most extensive set of connected authors".
198: "Figure 7 shows how the 15 authors with the greatest total link strength" -> correct grammar
215: Thresholds are given, but not really explained. Why do you select 50 as the minimum number of citations?
272: How were the clusters computed? Please explain in methods section.
296: Why is this table formatted differently than the others? The layout of the table is not self-explanatory... there seems to be information about the cluster always at the end of the cluster: "Label: Concept delimitation and accuracy of its measurement"... why "label"? Please consider repeating the column names of the table on later pages. The table does have inhomogeneous font sizes, e.g. numbers in cluster 2 are larger than in clusters 1 and 3.
288: Please explain, which "data clustering technique" has been used, as there are multiple algorithms and procedures for doing this. Maybe add a specific subsection to the methods section.
289: "The co-citations determine the correlation between the publications in terms of content." -> It is not clear, what is meant here.
291: Again, the methodology for choosing the thresholds is not clear. Why 20?
293: "were grouped in three clusters" -> The clustering algorithm and its parameters can highly affect the number of resulting clusters. Please comment on that.
305: Grammar
308: Grammar
DISCUSSION
316: "From 2021 onwards, the trend has been accelerated, with the most prolific years becoming the most productive." -> Where is this shown in this article.
318-324: Please correct grammar.
In parts, the discussion lists very specific results and in other parts it is very general (e.g. "Hence, this bibliometric study gives useful facts linked to research trends concerning the history, present, and future of working research."). I miss a discussion with respect to the concrete research question given in the introduction. Furthermore, the specific links between the results of this article and the research question are not detailed.
CONCLUSION
The conclusion is too generic. There is no real conclusion with respect to the results or the research question given in the introduction.
Author Response
I have reviewed and accepted the Associate Editor’s Decision, as noted below.
Response to 2nd Reviewer’s Comments
Reviewer’s Comment # 1: The abstract is concise and informative.
Authors’ Response to Comment # 1: Thank you for motivating appreciation.
Reviewer’s Comment # 2: The introduction is well-written, informative and cites many important articles. However, lines 44-66 are a mere listing without context and explanations. Context and links should be given and the paragraph might be shortened to give the most relevant articles.
Authors’ Response to Comment # 2: The said contents are linked according to your guidance. Furthermore, we shortened the paragraphs of antecedents and consequences of thriving. We added a Table 1 for the detailed contents.
Reviewer’s Comment # 3: 67: Not precise enough, what is meant exactly by "outcomes"?
Authors’ Response to Comment # 3: We tried our best to make it more clear and comprehensive.
Reviewer’s Comment # 4: 79: This is a strong statement which should have citations supporting it.
Authors’ Response to Comment # 4: Thank you for the constructive suggestion. We have cited the relevant work.
Reviewer’s Comment # 5: 95: "acknowledged" -> acknowledges?
Authors’ Response to Comment # 5: Narration error is corrected
Reviewer’s Comment # 6: 98: "the security of these themes" <- what is meant here?
Authors’ Response to Comment # 6: The confusion is eliminated in order to clear the stance.
Reviewer’s Comment # 7: 101: "The reset of the bibliometric mapping review is structured as follows." <- rest? remainder?
Authors’ Response to Comment # 7: The appropriate wording is utilized
Reviewer’s Comment # 8: 101-103: no verb
Authors’ Response to Comment # 8: The required verb is included in the sentence
Reviewer’s Comment # 9: 105: were -> are
Authors’ Response to Comment # 9: “Were” is replaced by “are”
Reviewer’s Comment # 10: The introduction is well-structured.
Authors’ Response to Comment # 10: Thanks for your appreciation
Method Section
Reviewer’s Comment # 11: 110: "mainstreams of study" -> studies?
Authors’ Response to Comment # 11: Correction has been made
Reviewer’s Comment # 12: 112: "presented in the figure 1"
Authors’ Response to Comment # 12: Correction has been made
Reviewer’s Comment # 13: 117: Caption should be expanded to describe the contents of the figure. Figure quality needs improvement (e.g. clipped text). Is the figure based on a (citable) standard procedure for bibliometric analyses? What do boxes mean, what is the difference between lines and arrows?
Authors’ Response to Comment # 13: Done. Caption and figure were adjusted and the quality was improved. The stages of the research process followed the bibliometric mapping research methodology. The arrows are related to the outcomes.
Reviewer’s Comment # 14: 131 Figure 2: Quality needs to be improved, figure seems squeezed. Caption needs a detailed description. Tick marks don't match data points. Why is the temporal resolution so low (5 years)?
Authors’ Response to Comment # 14: The quality of figure has been upgraded. In order to better explain the trend, 5 year temporal resolution has been applied. The caption and the figure are adjusted.
Reviewer’s Comment # 15: 133: Past and Present tense are mixed and should be standardized, e.g. "we conduct our second searching. Our search query was:" This should be unified throughout the whole article.
Authors’ Response to Comment # 15: A conscious effort has been made to unify the single sentence structure.
Reviewer’s Comment # 16: 138: "Figure 2 shows the five main areas and their percentage. As can be seen, management and business joined to achieve the 61% (Figure 3)." -> Fig. 3 instead of Fig. 2, I dont' understand the second sentence. Why were they joined?
Authors’ Response to Comment # 16: The correct figure number is assigned and blunders are eliminated.
Reviewer’s Comment # 16: 142: Caption needs to be corrected and expanded.
Authors’ Response to Comment # 16: The caption is improved
Reviewer’s Comment # 17: 144-146: Grammar and language needs to be streamlined. "more than what?" Information about the total number of countries is missing. The plot should be updated: A maximum value of 60 is sufficient for the x axis. Include a tick with a label between zero and 20. It is difficult to guess the numbers for the countries France to India.
Authors’ Response to Comment # 17: Figure is improved to make it more comprehensive
RESULTS
Reviewer’s Comment # 18: 156: Split sentence or correct grammar.
157: Using quotation marks or different font style would help to distinguish article titles from article text.
Authors’ Response to Comment # 18: Your effective comments are endorsed and obliged.
Reviewer’s Comment # 19: 164/171: Figure captions and table captions should always be on the same page as the respective figures.
Authors’ Response to Comment # 19: The required alignment has been made.
Reviewer’s Comment # 20: 171: Why is it table 3, where is table 2? Caption not detailed enough. Table formatting should be improved (line breaks etc.). Please explain "link strength" in methods section.
Authors’ Response to Comment # 20: Thanks for the suggestion. Suggestions followed.
Reviewer’s Comment # 21: 163: Figure 5 might contain very interesting information, but this figure is not interpretable in this form: graph layout needs to be improved for readability, fonts are too small, resolution of figure is not good enough. Figure caption needs to be expanded. Same for: Fig. 6-12. Why is everything in the graph figures in lower case? Please unify this to match the text.
Authors’ Response to Comment # 21: In the independent files, the resolution of the figures is well. The VOSViewer software provides the figures in lowercase.
Reviewer’s Comment # 22: Generally, the text flow in this section is often interrupted as figures, tables and text are entangled to heavily. Please restructure to have a more coherent look and feel of this section. Also the table formatting needs to be improved for all tables.
Authors’ Response to Comment # 22: Suggestions followed to make reading more linear
Reviewer’s Comment # 23: 188: "The total strength of co-authorship links with other authors was calculated." Please explain the concept of the "total strength of ... links" in the method section. Also, please add a few words about the "most extensive set of connected authors".
Authors’ Response to Comment # 23: Done. The paragraph is adjusted to give it more clarity. In the methodology section, the analysis is mentioned.
Reviewer’s Comment # 24: 198: "Figure 7 shows how the 15 authors with the greatest total link strength" -> correct grammar
Authors’ Response to Comment # 24: Correction has been made paragraph is adjusted to give it more clarity.
Reviewer’s Comment # 25: 215: Thresholds are given, but not really explained. Why do you select 50 as the minimum number of citations?
Authors’ Response to Comment # 25: We provide the justification for thresholds of 50 citations
Reviewer’s Comment # 26: 272: How were the clusters computed? Please explain in methods section.
Authors’ Response to Comment # 26: Done. Information included in the method section.
Reviewer’s Comment # 27: 296: Why is this table formatted differently than the others? The layout of the table is not self-explanatory... there seems to be information about the cluster always at the end of the cluster: "Label: Concept delimitation and accuracy of its measurement"... why "label"? Please consider repeating the column names of the table on later pages. The table does have inhomogeneous font sizes, e.g. numbers in cluster 2 are larger than in clusters 1 and 3.
Authors’ Response to Comment # 27: In order to better present the information, this kind of formatting was solution to avoid any ambiguity.
Reviewer’s Comment # 28: 288: Please explain, which "data clustering technique" has been used, as there are multiple algorithms and procedures for doing this. Maybe add a specific subsection to the methods section.
Authors’ Response to Comment # 28: The VOSviewer software provides the outcomes of the data clustering technique. There is no specific information about the algorithms.
Reviewer’s Comment # 29: 289: "The co-citations determine the correlation between the publications in terms of content." -> It is not clear, what is meant here.
Authors’ Response to Comment # 29: Done. We add some information to be clearer.
Reviewer’s Comment # 30: 291: Again, the methodology for choosing the thresholds is not clear. Why 20?
Authors’ Response to Comment # 30: We provide the justification for thresholds of 20
Reviewer’s Comment # 31: 293: "were grouped in three clusters" -> The clustering algorithm and its parameters can highly affect the number of resulting clusters. Please comment on that.
Authors’ Response to Comment # 31: The clusters were formed according to the published articles and according to the more realistic themes.
Reviewer’s Comment # 32: 305: Grammar
Authors’ Response to Comment # 32: Grammar mistakes are corrected.
Reviewer’s Comment # 33: 308: Grammar
Authors’ Response to Comment # 33: Grammar mistakes are corrected.
DISCUSSION
Reviewer’s Comment # 34: 316: "From 2021 onwards, the trend has been accelerated, with the most prolific years becoming the most productive." -> Where is this shown in this article.
318-324: Please correct grammar.
Authors’ Response to Comment # 34: The description of trend is corrected with the relevant year on trend line in the figure no 2.Morover, the erroneous sentences are improved.
Reviewer’s Comment # 35: In parts, the discussion lists very specific results and in other parts it is very general (e.g. "Hence, this bibliometric study gives useful facts linked to research trends concerning the history, present, and future of working research."). I miss a discussion with respect to the concrete research question given in the introduction. Furthermore, the specific links between the results of this article and the research question are not detailed.
Authors’ Response to Comment # 35: Thank you for this constructive comment. Your valuable comments have illuminated our thinking to make the necessary linkage between the research question and discussion.
CONCLUSION
Reviewer’s Comment # 36: The conclusion is too generic. There is no real conclusion with respect to the results or the research question given in the introduction.
Authors’ Response to Comment # 36: We endorse to oblige.
We want to thank the reviewer for the detailed comments that greatly helped us in the revision process. We appreciate the valuable input provided by reviewers and sincerely hope that the revised manuscript adequately addresses all the issues raised.
Round 2
Reviewer 2 Report
The authors did a very good job at improving the paper! All sections have been improved substantially.
Most of my previous comments have been addressed in sufficient detail.
I have a few comments which, I think, could improve the quality of the article:
Table 1:
Career satisfaction
and <- and?
lines 91 - 93
I suggest to at least add references to the VOSViewer manual (e.g. section 3.5.3 "analysis tab" includes information and references about clustering techniques) and also in the list
of publications there is a lot of material about clustering and community detection methods used
by the software that could be referenced: https://www.vosviewer.com/publications
I could not find a section explaining "link strength". You can find the corresponding explanation in
the VOSViewer manual:
Fig. 1:
The caption is understandable, but still, I find "related to the outcome" very brief. Could be improved.
137 please check lower/upper case of title
138-143 needs a grammar check
While the graphs in fig. 5-8 seem to be quite informative, I find it still hard to understand what information I'm actually supposed to
derive from fig. 9-11. The large number of edges in these networks makes it hard to understand
the network structure. Furthermore, I suspect that cluster/community analysis in these networks
is not very robust (meaning that few additional links or a slight parameter change is likely to
change the number and structure of clusters). Could you comment on that?
For fig. 9-11:
I think you should add information to the text and to the figure captions, what information is actually
contained in these figures and what results can be derived from these respective networks.
Just "Figure 11 shows the networks created between the authors according to bibliometric coupling analysis." is not really informative.
If you e.g. just want to visualize that the network is very densely coupled, please add a sentence like this to the article, together
with an interpretation of the respective outcome.
I'm especially confused about fig. 11: Is "hoefer, stefan" a special "node" - why is it aligned on the other end of the figure? What's it's difference to "rothmann, sebastiaan"?
269 grammar
294 "meta-analysis" or "meta-analyses"?
295 rather "studies in this cluster intend", right?
334-339 Could you add some references, where in the article this information is found?
404 Is there any input- or output- data for VOSViewer that you could make available?
Author Response
I have reviewed and accepted the Associate Editor’s Decision, as noted below.
Response to 2nd Reviewer’s Comments
Reviewer’s Comment # 1: The authors did a very good job at improving the paper! All sections have been improved substantially.
Most of my previous comments have been addressed in sufficient detail.
Authors’ Response to Comment # 1: Thank you for the appreciation.
Reviewer’s Comment # 2: Table 1: Career satisfaction and <- and?
Authors’ Response to Comment # 2: We deleted the word “and” in the revised document.
Reviewer’s Comment # 3: lines 91 - 93
I suggest to at least add references to the VOSViewer manual (e.g. section 3.5.3 "analysis tab" includes information and references about clustering techniques) and also in the list of publications there is a lot of material about clustering and community detection methods used by the software that could be referenced: https://www.vosviewer.com/publications
Authors’ Response to Comment # 3: We added another paragraph explaining the said information and reference of the manual.
Reviewer’s Comment # 4: I could not find a section explaining "link strength". You can find the corresponding explanation in the VOSViewer manual: https://www.vosviewer.com/documentation/Manual_VOSviewer_1.6.6.pdf
Authors’ Response to Comment # 4: Thank you for the constructive suggestion. We added paragraph in the method section explaining the link strength.
Reviewer’s Comment # 5: Fig. 1: The caption is understandable, but still, I find "related to the outcome" very brief. Could be improved.
Authors’ Response to Comment # 5: The outcome means the findings/results of that specific analysis. The confusion is eliminated in order to clear the stance.
Reviewer’s Comment # 6: 137 please check lower/upper case of title; 138-143 needs a grammar check
Authors’ Response to Comment # 6: Thank you. The whole paragraph is revised for grammar check.
Reviewer’s Comment # 7: While the graphs in fig. 5-8 seem to be quite informative, I find it still hard to understand what information I'm actually supposed to derive from fig. 9-11. The large number of edges in these networks makes it hard to understand the network structure. Furthermore, I suspect that cluster/community analysis in these networks is not very robust (meaning that few additional links or a slight parameter change is likely to change the number and structure of clusters). Could you comment on that?
For fig. 9-11: I think you should add information to the text and to the figure captions, what information is actually contained in these figures and what results can be derived from these respective networks.
Just "Figure 11 shows the networks created between the authors according to bibliometric coupling analysis." is not really informative.
If you e.g. just want to visualize that the network is very densely coupled, please add a sentence like this to the article, together with an interpretation of the respective outcome.
I'm especially confused about fig. 11: Is "hoefer, stefan" a special "node" - why is it aligned on the other end of the figure? What's it's difference to "rothmann, sebastiaan"?
Authors’ Response to Comment # 7: The figure 9-11 are added as the support of tables provided to further explain the link strength among the items. As mentioned previously, we added further explanation of the link strength.
Reviewer’s Comment # 8: 269 grammar; 294 "meta-analysis" or "meta-analyses"?
295 rather "studies in this cluster intend", right?
Authors’ Response to Comment # 8: Thank you for correcting.
Reviewer’s Comment # 9: 334-339 Could you add some references, where in the article this information is found?
Authors’ Response to Comment # 9: Thank you for the suggestion. Table numbers are added in the revised manuscript.
Reviewer’s Comment # 10: 404 Is there any input- or output- data for VOSViewer that you could make available?
Authors’ Response to Comment # 10: Sorry, it is not possible at the moment. However, we will try to upload after the approval.
We want to thank the reviewer for the detailed comments that greatly helped us in the revision process. We appreciate the valuable input provided by reviewers and sincerely hope that the revised manuscript adequately addresses all the issues raised.